# Application of Carbon-13 Isotopomer Analysis to Assess Perinatal Myocardial Glucose Metabolism in Sheep

**DOI:** 10.3390/metabo11010033

**Published:** 2021-01-05

**Authors:** Mukundan Ragavan, Mengchen Li, Anthony G. Giacalone, Charles E. Wood, Maureen Keller-Wood, Matthew E. Merritt

**Affiliations:** 1Department of Biochemistry and Molecular Biology, University of Florida, Gainesville, FL 32603, USA; mukundan@ufl.edu (M.R.); anthonygiacalone@ufl.edu (A.G.G.); 2Department of Pharmacodynamics, University of Florida, Gainesville, FL 32610, USA; mengchenli@ufl.edu (M.L.); kellerwd@cop.ufl.edu (M.K.-W.); 3Department of Physiology and Functional Genomics, University of Florida, Gainesville, FL 32603, USA; woodc@ufl.edu; 4Donald H Barron Reproductive and Perinatal Biology Research Program, University of Florida, Gainesville, FL 32603, USA

**Keywords:** isotopomer analysis, stable isotope tracer methodology, fluxomics, cardiac metabolism, fetal heart metabolism

## Abstract

Ovine models of pregnancy have been used extensively to study maternal–fetal interactions and have provided considerable insight into nutrient transfer to the fetus. Ovine models have also been utilized to study congenital heart diseases. In this work, we demonstrate a comprehensive assessment of heart function and metabolism using a perinatal model of heart function with the addition of a [U-^13^C]glucose as tracer to study central energy metabolism. Using nuclear magnetic resonance spectroscopy, and metabolic modelling, we estimate myocardial citric acid cycle turnover (normalized for oxygen consumption), substrate selection, and anaplerotic fluxes. This methodology can be applied to studying acute and chronic effects of hormonal signaling in future studies.

## 1. Introduction

The myocardium undergoes a transition from utilizing carbohydrates as the primary energy source during the perinatal period to using fatty acids preferentially as an adult [1,2]. In fetal lambs, prior to birth and for two weeks after birth, lactate is also an important source of energy. Lactate oxidation is, however, attenuated by fatty acids [3]. In tandem with metabolic changes, cardiomyocytes in fetal sheep lose their proliferation abilities under the process of terminal differentiation during the last thirty days of the gestation period [4]. Sheep fetal hearts are often used as a model for human congenital hearts since they share similarities in size, structural characteristics of myosin heavy chain, contractile kinetics, resting heart rates, and systolic and diastolic pressures [5].

Metabolic fluxes can often provide a more accurate assessment of the metabolic state of a biological system than metabolite concentrations. Stable isotope tracing methods have been developed to measure the distribution of isotopic enrichment by different analytical techniques such as mass spectrometry or nuclear magnetic resonance (NMR) spectroscopy. Carbon-13 NMR has excellent chemical selectivity and has been used to measure fluxes in pathways involved in central energy metabolism in perfused hearts [6,7,8], and livers [9,10]. More recently, carbon-13 NMR-based isotopomer analysis has also been used to study acetate metabolism in tumors in humans [11].

Ovine models of pregnancy have been previously used to study arterial-venous (A-V) differences in oxygen and glucose across the fetal heart [12]. These measurements have also been made in hypoxic conditions using the same models [13]. Although insightful, these studies do not provide granular information on substrate selection or fluxes through pathways involved in energy metabolism. In addition to the normal switch in substrate preference from carbohydrates to fatty acids, chronic maternal stress can cause changes in fetal myocardial metabolism [14]. Transcriptomic analyses showed differences in gene expression in several metabolic pathways between healthy and high-stress ovine models [14]. A method to directly study fluxes through the affected metabolic pathways would be highly beneficial to understand these changes and potentially chart a course of treatment. In this study, we have demonstrated the use of carbon-13 isotopomer analysis to directly study glucose metabolism in fetal sheep hearts. By infusing ^13^C enriched glucose and measuring oxygen consumed by the heart, we estimate citric acid cycle (TCA cycle) flux, and substrate selection in the fetal heart.

## 2. Results

The quality of the ^13^C NMR spectra from the fetal sheep heart samples facilitates a robust analysis of the metabolism (Figure 1). Qualitatively, the presence of D45 doublet and the quartet at C-4 of glutamate indicates extensive utilization of [U-^13^C]glucose, which produces doubly ^13^C labeled acetyl-CoA. Since the only ^13^C labeled substrate infused is [U-^13^C]glucose, interior carbons of glutamate can be ^13^C enriched either by pyruvate carboxylase flux (or malic enzyme [15]) or via several turns of TCA cycle. At the same time, ^13^C enrichment of interior carbons can be depleted due to anaplerotic flux through succinate or through cataplerotic pathways. The presence of an intense singlet at C-3 of glutamate is, therefore, depletion of ^13^C label at the interior carbons of glutamate, possibly due to active anaplerotic flux through succinate. Another measure of anaplerosis through succinate is the ratio of peak areas of C-4 to C-3 of glutamate (C4/C3 ratio). Consistent with the qualitative observation, the average C4/C3 ratios were 1.2 ± 0.14, 1.3 ± 0.23, and 1.2 ± 0.15, in the intraventricularseptum (IVS), left ventricle (LV), and right ventricle (RV), respectively.

Circulating glucose and lactate concentrations were measured prior to and during glucose infusion using catheters present in the brachial artery and coronary sinus. As expected, circulating concentration of glucose increased by about 30% with infusion of [U-^13^C]glucose, while lactate concentrations remained unchanged as shown in Figure 2a,b. Glucose and lactate consumption were, on average, 0.05 ± 0.001 mmoles/dl and 0.12 ± 0.003 mmoles/dl. Since two moles of lactate are derived from a single mole of glucose, total substrate utilized (as three carbon units) is 0.22 mmoles/dl. Of this substrate pool, ~0.034 mmoles/dl is ^13^C enriched (~34% increase multiplied with 0.10 mmoles/dl (1 mM) in terms of 3-carbon units), yielding an oxidative enrichment (i.e., acetyl CoA enrichment) of 15%. This enrichment readily matches the Fc3 (fraction of acetyl-CoA enriched at both carbons of acetyl group) values obtained from ^13^C spectra shown in Figure 2c.

Fetal myocardial oxygen consumption was measured in ten-minute intervals in order to assess function. As shown in Figure 3a, oxygen consumption was stable over the duration of infusion, indicating stability in myocardial energetics during infusion. Fitting the glutamate multiplets (Figure 1) to a minimal metabolic model yields constituent fluxes relative to citrate synthase. These relative fluxes when normalized to oxygen consumption yield absolute fluxes as shown in Figure 3b. Glucose and lactate oxidation are the major sources of acetyl-CoA, as evidenced by the differences in fluxes through pyruvate dehydrogenase and acetyl-CoA synthetase (Figure 3c). Interestingly, fetal hearts do not appear to discriminate between pyruvate and succinate anaplerosis, as shown in Figure 3d. It should be noted, however, that anaplerotic fluxes are about one-fifth of the TCA cycle flux.

With [U-^13^C]glucose as tracer, acetyl-CoA generated from glycolysis derived pyruvate will label C-4 and C-5 of α-ketoglutarate, which rapidly exchanges with glutamate via the glutamate dehydrogenase reaction. In the subsequent turns of the citric acid cycle, interior carbons of α-ketoglutarate (and consequently, glutamate) also acquire ^13^C label as shown in Figure 4. Here, ^13^C spectra of glutamate provide a read out of various fluxes contributing to central energy metabolism. In this study, the ^13^C tracer was infused for ~30 min in order to achieve metabolic steady state and provide measures of steady state TCA cycle flux. It is also possible to rapidly identify substrate selection under non-steady state conditions directly based on glutamate peak ratios [16]. However, this analysis cannot readily measure anaplerosis versus cataplerosis.

## 3. Discussion

A robust method to study cardiac metabolism of sheep hearts is of high utility since sheep hearts have similar resting cardiovascular parameters to humans [17] and are an excellent model of human myocardial metabolism. Since the methodology to measure oxygen consumption and blood flow is well established, carbon-13 isotopomer analysis can be readily integrated with existing methods to produce a quantitative assessment of myocardial energy metabolism. The inclusion of anaplerotic flux via succinate as a model component was necessitated by the notably lower ^13^C enrichment at the C-3 of glutamate versus the C-4. Pyruvate carboxylase flux alone cannot produce the necessary isotopomeric distributions, as the uniformly enriched carbon-13 labeled pyruvate precursor would maintain C3 labeling. Increased anaplerosis via succinate has been identified as upregulated in a model of heart failure [18]. It is to be determined if upregulated succinate anaplerosis is another manifestation of the return to the fetal metabolic phenotype associated with heart failure [19]. Using a single ^13^C tracer, it is clear that substrate selection and TCA cycle turnover can be obtained in a straightforward manner. In this study, the use of all three sections (LV, RV, and IVS) of heart allowed us to have replicate samples in each animal, but also to confirm that there are no differences in glucose metabolism between the chambers. Our previous metabolomic analysis of the fetal heart also did not find any differences in metabolites between these three sections of heart wall in newborns [20].

Perinatal myocardial metabolism on days 140–142 of sheep gestation is reported in this work. This time was chosen in order to reliably get fetal data prior to the onset of labor. Transcriptomic data [21] from hearts at this age of gestation suggest changes in cardiac metabolism in the model studied in our lab (chronic maternal stress/hypercortisolemia), and so this age was our starting point for establishing the stable isotope tracer method for use in subsequent studies. Additionally, since the ewes used in the study were not in labor yet, the observed metabolic fluxes are not labor contraction stimulated changes. Post-infusion, the arterial glucose increased from approximately 1.4 mM to 3 mM, and in the near-term fetus, it would be expected that this would cause additional insulin secretion (insulin concentration was, however, not measured), as the late gestation fetus has an increased insulin response to increases in glucose [22]. Since fetal glucose levels are increased normally during labor and delivery, the increase in fetal glucose is physiologically relevant.

In this work, we have studied glucose metabolism in perinatal hearts in healthy sheep. However, study of myocardial metabolism using this technique is not limited to these conditions. Stable isotope tracer methods can be readily applied to probing metabolic changes in response to biological factors including, but not limited to, intrauterine growth restriction. The general applicability of the stable isotope tracer approach is only limited by availability of robust measures of oxygen consumption needed to normalize fluxes obtained from analysis of ^13^C NMR spectra. In contrast with previous methods used to study metabolism in fetal hearts, the methodology reported here does not require the use of radioisotopes and provides information on not just fluxes but also metabolite concentrations (using either ^1^H NMR spectroscopy or other complementary techniques such as mass spectrometry). On the other hand, NMR spectroscopy is inherently insensitive and requires the use of ~150 mg of tissue to be able to record ^13^C spectra in less than 12 h of spectrometer time.

Perinatal hearts have been well known to prefer glucose as an energy source over fatty acids [12]. This substrate preference makes [U-^13^C]glucose a straightforward choice for stable isotope studies of metabolism. However, this does not limit the choice of tracers that can be utilized for probing more complex aspects of myocardial metabolism including response to treatments. Aided by measurements of circulating metabolite concentrations (e.g., glucose, lactate, amino acids, etc.), this approach to studying myocardial metabolism can also provide a means to interrogate associated pathways. In fact, further studies with a mixture of ^13^C tracers and targeted metabolomics in an ovine model of maternal stress is already underway in our groups.

## 4. Materials and Methods

### 4.1. Animal Use

Time-dated pregnant ewes (*n* = 5) were studied on days 140–142 of gestation. Ewes were fasted overnight before surgery. A jugular venous catheter was placed using aseptic technique, and anesthesia was induced with intravenous propofol and ketamine (5–7 mg/kg propofol, 1.5 mg/kg ketamine). Ewes were intubated, and anesthesia was maintained during surgery with a continuous infusion of propofol (18–30 mL/kg/h) and ketamine (3–6 mg/kg/h) through a jugular venous catheter; ventilation was controlled during the surgical procedure. The blood pressure, heart rate, ECG, SPO_2_, and end-tidal CO_2_ of the ewe were monitored throughout the procedure. Catheters were placed in the fetal tibial artery and advanced to the inferior vena cava for infusion of [U-^13^C] glucose. Catheters were placed in the fetal brachial artery and in the coronary sinus via the hemiazygos vein for sampling of blood to determine arterial-venous (av) differences in glucose and lactate across the fetal heart. Sampling was carried out approximately every 10 min. Prior to the start of the [U-^13^C]glucose infusion, microspheres (BioPal Inc, Worcester, MA, USA) were injected into the inferior vena cava to measure blood flow. A reference sample was collected over 3–3.5 min from the aorta via the catheter placed in the brachial artery. [U-^13^C] glucose (Cambridge Isotope Laboratories, Inc., Andover, MA, USA) was infused into the vena cava for 30 min (0.3–0.4 g [U-^13^C]glucose/kg) to achieve steady state levels of glucose in the arterial blood. Samples were collected prior to the start of the infusion and during the infusion. Fetal P_O2_ was determined in both arterial and coronary venous samples (Radiometer ABL 80 Flex Plus, Radiometer Inc, Brea, CA, USA). At the end of the infusion, euthanasia solution (Euthasol, Virbac AH, Inc., Fort Worth, TX, USA; ~5.8 g pentobarbital sodium and 0.75 g phenytoin sodium) was administered to the ewe and the fetal heart was rapidly collected. A transverse section of the fetal heart was rapidly frozen in liquid nitrogen; dissected into right ventricular free wall, left ventricular free wall, and intraventricular septum; and frozen sections placed in tubes. The tubes were placed in liquid nitrogen and then stored at −80 °C until analysis. Additional samples of the heart were collected for microsphere quantification. Aliquots of plasma were frozen, and plasma glucose and lactate concentrations were determined (YSI model 2700 Glucose/Lactate Analyzer, Yellow Springs Life Sciences, Yellow Springs, OH, USA). The use of animals in this study was approved by the University of Florida Institutional Animal Care and Use committee (protocol# 201909496).

Blood flow to the heart was estimated using the microsphere technique [23,24] in four of the fetuses. Samples of the heart and the arterial reference sample were sent to BioPal for analysis. This method uses injections of nonradioactive microspheres (50 µm; Iridium, Samarium and Luteum were used in this study), which are activated to their radioactive isotopes for determination of numbers in the arterial reference sample and in the cardiac sections. Cardiac blood flow in ml/100 mg tissue by comparison of disintegrations per minute (dpm) in the cardiac sample relative to dpm in the arterial reference sample:(1)$$\mathrm{Flow}\;\left(Q\right)=\;\frac{\mathrm{flow}\;\mathrm{in}\;\mathrm{reference}\;\mathrm{sample}\;\left(\frac{\mathrm{mL}}{\min}\right)\times \;\mathrm{dpm}\;\mathrm{in}\;100\;\mathrm{mg}\;\mathrm{heart}\;}{\mathrm{dpm}\;\mathrm{in}\;\mathrm{reference}\;\mathrm{sample}}$$

### 4.2. Extraction of Metabolites

Frozen sections (intraventricular septum (IVS), left ventricle (LV), and right ventricle (RV); ~250–400 mg of tissue) of the heart were homogenized in a Fastprep 24 homogenizer (MP Biomedicals, Irvine, CA, USA) using a solvent system containing acetonitrile, isopropanol, and water mixture (volume ratio of 3:3:2). Homogenized tissues were centrifuged at 10,000× *g* and 4 °C for 30 min to remove tissue debris. Supernatant was lyophilized in a Speedvac system (Thermo Scientific, Waltham, MA, USA) overnight. The dry powder was dissolved in a 1:1 mixture of acetonitrile and water and centrifuged as before to remove excess salt from the sample. Supernatant was lyophilized overnight and stored at −20 °C.

Samples for nuclear magnetic resonance spectroscopy were prepared by dissolving the lyophilized powder in D_2_O containing 50 mM sodium phosphate, pH 7.0, and 2 mM ethylene diamine tetraacetic acid (EDTA). To this solution, a 10% (*v/v*) standard solution containing 5 mM 3-(Trimethylsilyl)-1-propanesulfonic acid-d6 (d6-DSS) and 0.2% (*w/v*) sodium azide was added. The sample was centrifuged for 15 min at 4 °C and 10,000× *g*, and supernatant transferred into 1.5 mm capillary NMR tube (New Era Enterprises, Newfield, NJ, USA) using a 50 µL glass syringe (Hamilton, NV, USA). Total sample volume was ~55 µL.

### 4.3. NMR Spectra Acquisition

^13^C spectra were acquired using a 14.1 T magnet running an NMR spectrometer (Agilent Technologies, Santa Clara, CA, USA) equipped with a 1.5 mm home-built high temperature superconducting (HTS) probe [25]. Spectra were acquired using a 45° radiofrequency pulse with an acquisition time of 1.5 s, pre-scan delay of 1.5 s, spectral width of 244 ppm (55147 complex points), and ^1^H decoupling (decoupling field strength of 4800 Hz; WALTZ-16).

^1^H spectra were acquired using a 18.8 T magnet running an NMR spectrometer (Bruker Biospin, Billerica, MA, USA) equipped with a 5 mm TCI cryoprobe (Bruker Biospin). ^1^H spectra were acquired with a spectral width of 15 ppm (8192 complex points), and an acquisition time of 0.68 s in an interleaved manner with and without ^13^C decoupling (decoupling field strength of 2100 Hz; garp4).

### 4.4. Steady State Analysis

^13^C spectra were zero filled to 131,072 points, line broadened using a 0.8–1 Hz exponential decay, and baseline corrected. ^1^H spectra were zero-filled to 131,072 points and 0.35 Hz exponential line broadening. Baseline correction was carried out using a fifth order polynomial function. All spectra were processed using MestReNova v14.1 (Mestrelab, Spain). ^13^C chemical shifts were calibrated using the singlet resonance of taurine (set to 48.4 ppm).

Glutamate resonances were fit to a mixed Gaussian/Lorentzian shape to obtain peak areas. These ^13^C peak areas representing isotopomer abundance (along with ^13^C enrichment of alanine obtained from ^1^H spectra) were provided as input to a minimal metabolic model (Table 1) and solved numerically using tcaCALC [26]. tcaCALC performs a least-squares fitting procedure yielding best estimates of relevant fluxes. Relative fluxes obtained from the model fit were converted to absolute fluxes using oxygen consumption values and appropriate weighting factors [27] reported in the literature.

## Figures and Tables

**Figure 1 metabolites-11-00033-f001:**
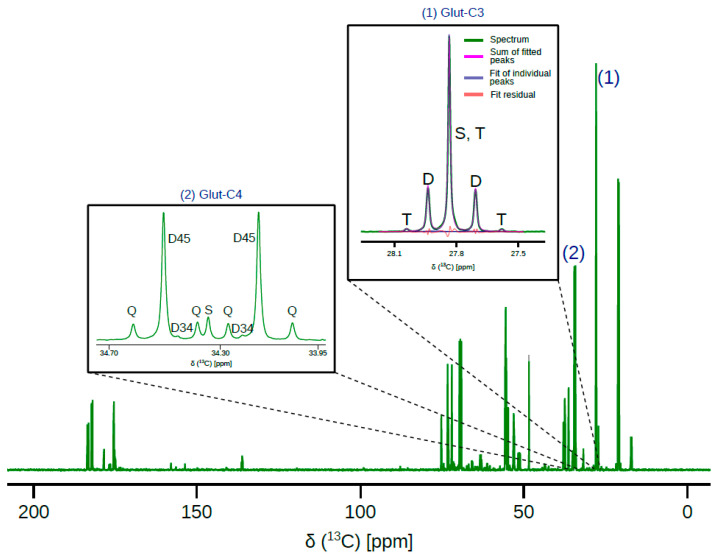
Representative ^13^C NMR spectrum of metabolites extracted from a fetal sheep heart right ventricle. Inset (1) shows ^13^C resonance of C-3 of glutamate along with line fitting carried out to estimate peak ratios. Inset (2) shows ^13^C resonance corresponding to C-4 of glutamate. Line fitting is not shown in inset (2) so as to not obscure the D34 doublet. In inset (1), “T” represents triplet arising from ^13^C label in carbons 2-3-4 of glutamate, “D” represents 2-3 and 3-4 isotopomers, and “S” arises ^13^C label in carbon-3 of glutamate in addition to natural abundance. In inset (2), “Q” represents quartet arising from ^13^C label in carbons 3-4-5 of glutamate; “D45”, doublet from 4-5 of glutamate; “D34”, doublet from 3-4 of glutamate; and “S” is singlet from ^13^C label in carbon-4 of glutamate in addition to natural abundance. ^13^C spectra acquisition parameters are detailed in Section 4.

**Figure 2 metabolites-11-00033-f002:**
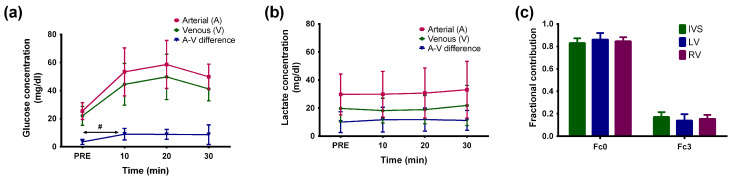
Arterial, venous, and arterial-venous (A-V) differences in (**a**) glucose and (**b**) lactate (right) concentrations prior to (PRE) and during the course of [U-^13^C] glucose infusion. Error bars are standard deviation (*n* = 5; 2 males and 3 females). **#** indicates statistically significance (“PRE” vs. “10 min.”; *p* < 0.05; two–tailed Student’s *t*–test). Glucose and lactate concentrations were measured as described in *Methods* section. (**c**) Fractions of acetyl-CoA without ^13^C enrichment (Fc0) and enriched at both carbons of the acetyl group (Fc3) estimated from ^13^C NMR spectra (Figure 1).

**Figure 3 metabolites-11-00033-f003:**
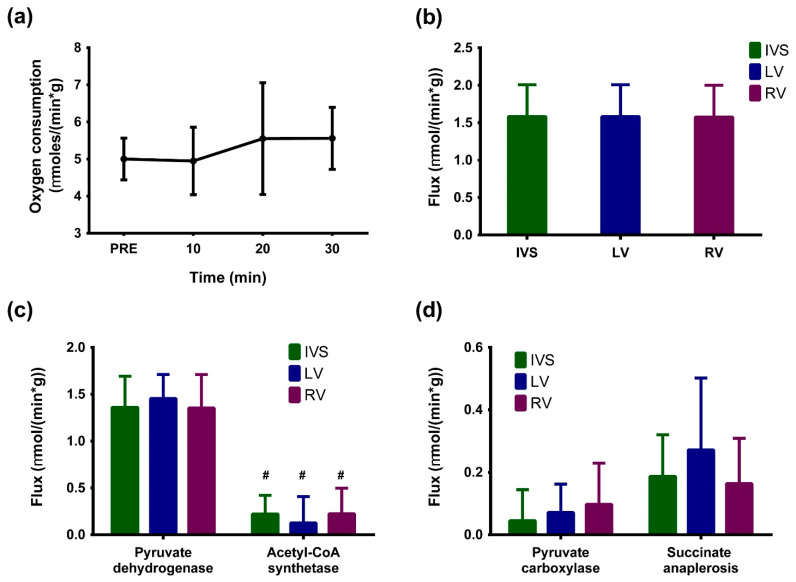
(**a**) Myocardial oxygen consumption measured during the course of [U-^13^C] glucose infusion (*n* = 4). “PRE” refers to myocardial oxygen consumption prior to start of infusion. (**b**) Citric acid cycle turnover in intraventricular septum (IVS), left ventricle (LV), and right ventricle (RV) sections of fetal heart. Oxygen consumption averaged over 30 min is used to obtain absolute citric acid cycle flux. (**c**) Fluxes through pyruvate dehydrogenase (PDH) and acetyl-CoA synthetase (ACS) representing sources of acetyl-CoA in different sections of the fetal heart. **#** indicates statistical significance (PDH vs. ACS for each section of the heart; *p* < 0.001; two–tailed Student’s *t*–test) (**d**) Fluxes through anaplerotic pathways in different sections of the fetal heart. A minimal metabolic model including explicit definitions of pyruvate dehydrogenase flux, pyruvate carboxylase flux, and succinate anaplerosis was used to fit the data and obtain estimates shown in (**b**–**d**). Error bars in all panels are standard deviation (*n* = 5; 2 males and 3 females).

**Figure 4 metabolites-11-00033-f004:**
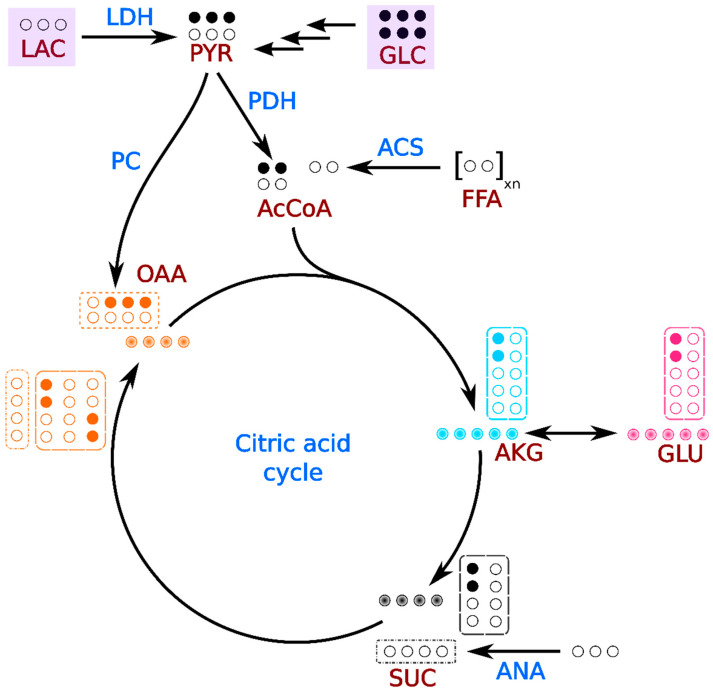
Pathway diagram showing the propagation of ^13^C label through citric acid cycle and anaplerotic intermediates. Solid-filled circles represent ^13^C label and gradient-filled circles represent all possible isotopomers for each metabolite. Isotopomers generated in the first turn of citric acid cycle and anaplerotic fluxes are indicated by dashed and dot-dashed boxes respectively. Carbons are numbered in right-to-left and bottom-to-top directions. Multiple arrows indicate multi-step pathways (such as glycolysis). Metabolites are abbreviated as follows: GLC: glucose, PYR: pyruvate, LAC: lactate, AcCoA: acetyl-CoA, FFA: free fatty acids, OAA: oxaloacetate, AKG: α-ketoglutarate, GLU: glutamate, and SUC: succinate. Fluxes are abbreviated as follows: LDH: lactate dehydrogenase, PDH: pyruvate dehydrogenase, PC: pyruvate carboxylase, ACS: acetyl-CoA synthetase, and ANA: anaplerosis through succinate.

**Table 1 metabolites-11-00033-t001:** Starting values and acceptable ranges (in parenthesis) of parameters used in metabolic modeling. ^13^C enrichment of lactate is obtained from the methyl resonance. Starting values represent magnitude of fluxes relative to unitary citrate synthase flux.

Parameter	Explanation	Starting Value (Range)
lac123	^13^C enrichment of lactate (derived from [U-^13^C]glucose	Data from ^1^H NMR spectra
pdh	Pyruvate dehydrogenase flux	0.3 (0–1)
ys	Anaplerosis through succinate	0.3 (>= 0)
ypc	Pyruvate carboxylase	0.05 (>= 0)

## Data Availability

Not applicable.

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
