# Peer review of "Application of Carbon-13 Isotopomer Analysis to Assess Perinatal Myocardial Glucose Metabolism in Sheep"

_metabolites, 2021, doi:10.3390/metabo11010033_

Round 1

Reviewer 1 Report

This study uses continuous infusion of 13-labeled glucose to fetal sheep hearts in order to assess arterial and venus blood glucose/lactate levels. Cardiac tissue samples were then collected and analyzed using 13C-NMR spectrometry at several time points to estimate labeling of glucose/glycolysis-related metabolites and calculate metabolic pathway flux, normalized to O2 consumption. As a ‘proof of concept’, the paper is OK. I wish there was a more explicit hypothesis being tested (but I am an experimentalist at heart). That said, there are important bits of information missing, especially from the methods section (e.g. info. about animal numbers, etc.).

Major Comments:

The last line of the abstract states that this “methodology can be used to study…hormonal signaling” but there is no discussion of this at all in the rest of the paper. Generally, the brief and doesn’t really expand on potential future uses, besides alluding (quite briefly) to their potential.

The introduction is quite brief. While it provides a bit of background on the technique to be employed in the study, it poses no real biological questions or hypotheses. If this is, after all, to be a purely descriptive study, then I would appreciate more motivation (expand on why this approach could be useful, compared to previously used techniques). Alternatively, are the authors assessing the suitability of this technique for addressing questions in their subfield? Then perhaps more explicitly state that as a study goal.

The methods section is missing a great deal of important information. For example, nowhere do I see sample sizes presented. Neither do I see a clear incorporation of the temporal sampling regime into the methods description. As far as I can determine, no details regarding the “metabolic modelling” done are presented in the methods section and only a reference/computer package is listed. I would appreciate at least a brief description of how this modelling works.  

Other concerns:

Title: I’m not sure the qualifier “complete” is warranted. The authors themselves acknowledge that they cannot use the identified methodology to distinguish anaplerotic from cataplerotic flux.

Line 48 – “Outstanding” is a subjective term. And “The quality of the…” reads just fine.

Line 54 – Insert “an” between “of” and “intense”.

Line 58 – These abbreviations are not defined prior to their first use.

Line 61 – Change “spectrum” to “spectra”.

Line 189 – “right ventrivle” is misspelled.

Line 200 – Change “is” to “was” to maintain consistency.

Figures – A fair amount of the text in the figure keys is VERY small and difficult to read.

Author Response

Please see the attached response.

Reviewer 2 Report

It is a shame that no comparative assessments were made post birth or that fatty acids were not assessed. However despite this, the paper is still provides valuable information and the authors should be commended on successfully undertaking a technically challenging study that is of high significance. I have a few minor comments.

- Can the authors specify whether their results primarily represent contraction or insulin stimulated cardiac glucose metabolism? Is there a large insulin response post U- 13C glucose infusion at 0.3-0.4g/kg?

- The authors assessed cardiac metabolism at just one stage during foetal development (140-142 days into gestation). Can the authors comment on the time chosen during gestation in relation to maturation of cardiac energy metabolism during later state gestation/post birth.

- The authors compared LV, RV and IVS, however it is not clear why the authors assessed glucose metabolism between the different cardiac sections. Are there differences in cardiac energy demands between these sections at this stage of development?

- Do the authors have maternal plasma glucose/insulin data?

- And are there sex differences between the foetal hearts assessed?

Reviewer 3 Report

Ragavan et al. examined the use of [U-13C]glucose labeling as a means to determine glucose metabolism in fetal hearts of sheep. While the figures and results of this manuscript are fine, the title of this manuscript confuses the reader as to what this manuscript is; a methods paper. This manuscript is showing that [U-13C]glucose tracing can be used as a tool to examine maternal-fetal interactions and glucose metabolism in the hearts of offspring. My comments for this manuscript are minor.

  1. The title doesn't seem to reflect the manuscript: this is a methods paper showing that this technique can be used to examine nutrient metabolism.
  2. The authors briefly touch on other tools in the introduction compared to this method. However, in the discussion it would have been nice to know the advantages to this system compared to existing methods as well as disadvantages.
  3. While not in the scope of this manuscript, it would also have been interesting to know if this method would be useful in response to stressors like intrauterine growth restriction, etc. 
